# Are We Closer to International Consensus on the Term ‘Food Literacy’? A Systematic Scoping Review of Its Use in the Academic Literature (1998–2019)

**DOI:** 10.3390/nu13062006

**Published:** 2021-06-10

**Authors:** Courtney Thompson, Jean Adams, Helen Anna Vidgen

**Affiliations:** 1School of Exercise and Nutrition Sciences, Faculty of Health, Queensland University of Technology (QUT), Victoria Park Road, Kelvin Grove, QLD 4059, Australia; h.vidgen@qut.edu.au; 2Centre for Diet and Activity Research (CEDAR), MRC Epidemiology Unit, University of Cambridge School of Clinical Medicine, Cambridge CB2 0SL, UK; jma79@medschl.cam.ac.uk

**Keywords:** food literacy, systematic scoping review, definition, concepts, application

## Abstract

(1) Background: The term ‘food literacy’ has gained momentum globally; however, a lack of clarity around its definition has resulted in inconsistencies in use of the term. Therefore, the objective was to conduct a systematic scoping review to describe the use, reach, application and definitions of the term ‘food literacy’ over time. (2) Methods: A search was conducted using the PRISMA-ScR guidelines in seven research databases without any date limitations up to 31 December 2019, searching simply for use of the term ‘food literacy’. (3) Results: Five hundred and forty-nine studies were included. The term ‘food literacy’ was used once in 243 articles (44%) and mentioned by researchers working in 41 countries. Original research was the most common article type (*n* = 429, 78%). Food literacy was published across 72 In Cites disciplines, with 456 (83%) articles from the last 5 years. In articles about food literacy (*n* = 82, 15%), review articles were twice as prevalent compared to the total number of articles (*n* = 10, 12% vs. *n* = 32, 6%). Fifty-one different definitions of food literacy were cited. (4) Conclusions: ‘Food literacy’ has been used frequently and broadly across differing article types and disciplines in academic literature internationally. However, agreement on a standardised definition of food literacy endorsed by a peak international agency is needed in order to progress the field.

## 1. Introduction

While many aspects of the food system, such as availability, accessibility, price and affordability, have been explored and evaluated, there is a limited understanding of the relationship between these factors and people’s food acquisition and consumption. The term ‘food literacy’ emerged in contemporary nutrition policies and plans as early as 1990 and in published literature from 2001 as the everyday skills, behaviour and knowledge needed by individuals to navigate the food environment and meet their nutrition and health needs [1,2,3].

The term ‘food literacy’ has been used by industry, public health nutrition organisations and policy-makers to encompass anything from food preparation to cooking skills, food science, household food production, food safety and food marketing [4]. It has gained momentum globally and been used to inform the development of questionnaires, observational studies and interventions by researchers in Australia [5], Italy [6], the Netherlands [7], South Africa [8], Switzerland [9], France [10], the United Kingdom (UK) [11] and the United States of America (USA) [12,13]. In 2017, Truman and colleagues [4] reported 38 novel definitions of food literacy. There are also numerous frameworks and models proposing relationships between food literacy and various food-related outcomes such as diet quality, nutrition behaviours, social connectedness and food security [14,15]. Despite food literacy having relevance across a broad range of countries and contexts, a lack of clarity around its definition, conceptualisation and operationalisation has resulted in inconsistencies in food literacy research. For example, current questionnaires developed to measure food literacy vary substantially, even among those citing the same definition and conceptualisation [16]. This lack of shared understanding of food literacy inhibits the synthesis of findings and limits the potential for leveraging food literacy to improve dietary intake and food security. Previous researchers have conducted reviews attempting to reach consensus on the definition and attributes of food literacy. However, limitations to Western countries [17], inclusion of a conceptualisation [4] or year of publication restrictions [18] means that the full breadth, reach and scope of the term ‘food literacy’ has yet to be explored.

Therefore, the present research goes beyond the existing works by looking broadly at the use of the term ‘food literacy’ throughout the whole peer-reviewed literature, regardless of context, to better understand its reach and application. Additionally, this article aimed to determine if ‘food literacy’ is a widely used and understood term to explore the potential for an internationally endorsed definition of food literacy. The objectives were to conduct a systematic scoping review to (1) describe frequency of the use and reach of the term ‘food literacy’, (2) identify changes in the use of the term over time and (3) describe how the term has been applied and defined within the literature.

## 2. Materials and Methods

This scoping review was planned and conducted using the Preferred Reporting Items for Systematic Reviews and Meta-Analyses: Extension for Scoping Reviews (PRISMA-ScR) guidelines [19].

### 2.1. Search Strategy

A systematic literature search was undertaken to identify peer-reviewed journal articles that used the term food literacy in their main text. The search term was simply ‘food literacy’ entered into the all fields or all articles search in the following databases: PubMed, ScienceDirect, Embase, Scopus, EBSCO (CINHAL + Medline), ProQuest and Google Scholar up to the 31st of December 2019. No date limitations were applied, and the term could appear anywhere in the text, excluding the reference list or footnotes.

### 2.2. Screening and Selection

Articles obtained from the search strategy were imported into EndNote [20], and duplicates were removed. The inclusion and exclusion criteria were applied at two stages of the review. In phase one, the abstract and title of the records were screened and excluded if they were: (1) not in English or a (2) book, (3) thesis or (4) not published in a peer-reviewed journal. In phase two, the full texts of articles were screened and excluded if: (1) they met the exclusion criteria of phase one, (2) the full text was not locally available or provided by authors via ResearchGate, (3) if the term ‘food literacy’ was not used and (4) if the term ‘food literacy’ was only found in the reference list or footnote of the article.

### 2.3. Data Extraction

To describe the frequency of use and reach of the term, the following data was extracted from each article:(1).The number of times the term ‘food literacy’ appeared within the main text (i.e., title, abstract, body, tables and figures) was indicative of the authors’ scale of understanding [21,22,23,24]. Neuendorf [25] proposed using variables to categorise the data, whereby Rossi and Macagno’s [26] three-point scale of understanding was used: (i) weak understanding (articles that used the term ‘food literacy’ infrequently or inconsistently in varying contexts); (ii) acceptable understanding (articles that used the term ‘food literacy’ somewhat frequently, consistently and in relevant contexts) or (iii) strong understanding (articles that used the term ‘food literacy’ frequently and consistently in a relevant context) [21].(2).The country of affiliation of the first author of articles, categorised by: (i) continent and (ii) World Bank income group [27].(3).The article type, determined by the journal’s classification of the type of article and categorised as: (i) original research, (ii) review articles, (iii) perspectives, (iv) short reports and communications, (v) case studies and (vi) unspecified.(4).The discipline, defined according to the journal’s subject categories in In Cites, the Clarivate Analytics journal comparison database [28].

To describe the use of the term over time, the year of articles were extracted and cross-tabulated against the total number of articles, country of affiliation of the first authors of articles and the discipline of articles.

To describe how food literacy has been applied and defined, articles with a primary focus on food literacy, determined by the use of the term in the title, were collated and compared across article types [29]. The following information was also extracted:(1).Aims of the article, stated in either the abstract or introduction and categorised into: (i) expert opinion articles, (ii) definition articles, (iii) intervention/program articles, (iv) measurement articles and (v) observational articles.(2).The definition of food literacy, determined by either a direct quote or the first citation after the term ‘food literacy’ was first used, where differing versions of definitions attributed to the same source were combined. To compare the content of the most-cited definitions, a thematic analysis of the definitions conducted by Truman et al. [4] into the following six categories were reported here: (i) skills/behaviours (physical actions/abilities involving food), (ii) food/health choices (actions associated with informed choices around food use), (iii) culture (societal aspects of food), (iv) knowledge (ability to understand and seek information about food), (v) emotions (attitudes and motivation) and (vi) food systems (complexity of food systems, including environmental impact and food waste).

### 2.4. Data Analysis

All three authors independently reviewed the same random samples of 50 articles and applied the criteria in phases one and two. Discrepancies were discussed and the methods revised to reduce these. C.T. then extracted the data from all the articles using the revised methods. Data were charted and analysed using descriptive statistics.

## 3. Results

The search strategy identified 2086 unique records (see Figure 1). Five hundred and forty-nine articles met the inclusion criteria and were included in the subsequent analysis (see Appendix A).

### 3.1. Frequency of Use and Reach of the Term

Refer to Appendix A, tab 3.1a to 3.1d for the article analysis of this section.

The distribution of the frequency of the use of ‘food literacy’ within the articles was skewed (range 1–189, median = 2). The term ‘food literacy’ was used once in 243 of the 549 articles (44%) and between two and five times in 160 of the 549 articles (29%) (see Figure 2). Articles that suppressed the understanding of the construct by using the term infrequently, inconsistently and in varying contexts cited ‘food literacy’ between one and five times (*n* = 403, 73%). Some understanding of the construct and consistency in the frequency and relevance was identified in the articles that used the term between 6 and 43 times (*n* = 101, 18%), while the articles that were considered to improve understanding of the construct used the term more than 44 times in relevant and appropriate contexts (*n* = 45, 8%).

The term ‘food literacy’ was mentioned in academic literature by the first authors working in 41 countries (see Table A1) across all continents. The term was used most frequently by teams where the first author was located in Australia (*n* = 127), Canada (*n* = 116), the United States of America (*n* = 112), the United Kingdom (*n* = 37) and Italy (*n* = 18) (see Figure A1). Additionally, there was a spread of first authors from lower-middle income countries (*n* = 5), upper-middle income countries (*n* = 11) and high-income countries (*n* = 25) [27,31]. No first authors were working in low-income countries.

Original research was the most common article type (*n* = 429) and accounted for 78% of all the articles using the term ‘food literacy’. There were 32 review articles published using the term ‘food literacy’ (6%); 27 perspectives, opinions or commentary pieces (5%) and 18 short reports or communications (3%). Case studies were the least common article type (*n* = 10, 2%). Thirty-three (6%) articles were not classified.

The term ‘food literacy’ was used in journals representing 72 of the 235 In Cites disciplines. These included agriculture, business, medicine, economics, education, environmental science, geography, hospitality, psychology/psychiatry, sociology and sports sciences. Overall, the term was used most frequently used in disciplines such as ‘public, environmental and occupational health’ and ‘nutrition and dietetics’ (see Table A2). The journals most frequently represented in these disciplines were *Nutrients* (*n* = 27), *Public Health Nutrition* (*n* = 16), *Appetite* (*n* = 16), *Canadian Journal of Dietetic Practice and Research* (*n* = 15) and *Journal of Nutrition Education and Behavior* (*n* = 14).

### 3.2. Use of Term over Time

Refer to Appendix A, tab 3.2a to 3.2d for the article analysis of this section.

The number of articles using the term ‘food literacy’ has increased over time (see Figure 3). Overall, 83% of articles using the term ‘food literacy’ occurred in the five years preceding this search (2015–2019).

The authors based in the USA were the first to publish on food literacy in 1998 (see Figure 4), with the authors based in North America the only continent represented between 1998 and 2004. The earliest article from a first author based in a non-English-speaking country, Italy, occurred in 2007, and the earliest article with a first author based in a low-middle income country came from Nigeria in 2011. There were no further works from first authors based in low-middle income countries until 2018.

The proportion of articles in each article type increased yearly, with the highest number of original research (*n* = 138), short reports (*n* = 7), review articles (*n* = 11), perspectives (*n* = 9) and case studies (*n* = 2) reported in 2019 (see Figure 5). Overall, 33 articles were ‘not classified’.

The number of disciplines represented each year, with the most frequent disciplines highlighted, can be seen in Figure 6. The first article on food literacy from 1998 was in the discipline of ‘cardiac and cardiovascular systems’. However, overall, the discipline of ‘nutrition and dietetics’ was the most highly represented (*n* = 136, 17%).

### 3.3. Applications and Definitions of Food Literacy

Eighty-two articles (15%) used the term ‘food literacy’ in the title of the article. The number of times the term was used throughout these articles ranged from 6–189 (refer to Appendix A tab 3.3a to 3.3d for the article analysis of this section).

The distributions of article types between all the articles on food literacy (*n* = 549) and those primarily about food literacy (*n* = 82) were similar for most categories, with the exception of original research and review articles (see Table 1). The number of articles classified as original research was 13 percentage points less in articles primarily about food literacy compared to the total number of articles, while review articles were twice as prevalent in articles primarily about food literacy (12% vs. 6%).

The aim of articles with a primary focus on food literacy can be seen in Table 2. Over half of articles primarily about food literacy were either opinion pieces or definition articles exclusively using experts as their research subjects (62%, *n* = 51), while 21% (*n* = 17) used participants exclusively from the general public. Observational and measurement articles had a combination of expert and general public involvement.

The definitions used by articles when referencing food literacy are described in Table A3. Overall, there were 51 different definitions of food literacy cited by the authors of the 82 articles considered to be ‘about’ food literacy. The most frequently cited definition of food literacy was that of Vidgen and Gallegos (2014) [15] (*n* = 66, 41%). Other commonly cited definitions were Cullen et al. (2015) [18] with 12 citations (7%), Kolasa et al. (2001) [2] with 7 citations (4%) and Verlado (2015) [32] with 5 citations (3%). Thirty-three of the definitions were only cited once. Of the four definitions most frequently cited, none addressed all six categories of skills/behaviours, food/health choices, culture, knowledge, emotions and food systems (Table A3). The most comprehensive definitions met five categories [15,18], while the least only addressed two [2,32].

## 4. Discussion

The purpose of this study was to better understand the reach and application of the term ‘food literacy’ in order to progress the field. This study found that, while the term ‘food literacy’ has been used frequently and broadly throughout academic literature, there are inconsistencies in its application and definition.

### 4.1. Use of the Term

The term ‘food literacy’ is widely used and has been described in a variety of different research disciplines and plethora of contexts. This is not limited to just ‘food’, usually discussed in the context of nutrition and dietetics or public health, but also covers the ‘literacy’ aspect of the construct, within disciplines such as education, communication, literature, language and linguistics. Begley and Vidgen [3] indicated that this may reflect attempts by a range of food-related sectors to describe the totality of food and eating as opposed to focusing on the singular issue of maximising dietary quality for good health. The term has clearly resonated with researchers with a very wide range of interests, indicating some level of consensus that translates to interventional value.

The number of articles using the term ‘food literacy’ has increased over time. A substantial number of food literacy articles in the latter five years of our data collection period align with the emergence of key articles defining (2014–2017) [4,15,17,18], conceptualising (2016) [33] or measuring (2017 and 2018) [5,6,7,9] the construct. In 2016, Begley and Vidgen proposed that the interest in food literacy was driven by the increasing prevalence of diet-related disease and a recognition that contemporary nutrition science needs to look beyond the biological determinants [3]. While this is still the case, food literacy is now also discussed more broadly in the context of food environments and food security [7,17,34,35,36]. Additionally, the type of research on food literacy has increased across all categories over time. While the first few articles on food literacy were original research articles, subsequent initial articles were perspectives articles, and most reviews were published in the last six years. This progression is typical in fields of research, as review genres evaluate knowledge claims and draw on theories and methods to enhance their credibility [37].

The term ‘food literacy’ has been used in both national and international contexts, in both English and non-English-speaking countries of differing income statuses, highlighting the broad reach of the term. This is promising for international consensus of the construct. However, while the term appears to be used frequently throughout the literature, just under half of the articles included in this review cited the term ‘food literacy’ only once in the manuscript. According to Cukier and colleagues [21], the number of times a term appears provides insight into themes that dominate the discourse, and omissions that may suppress understanding. Given that majority of articles used the term so infrequently and, in some cases, just as a keyword suggests that it is often used as an indicator of a general topic area rather than a point of particular depth. This can make navigating food literacy literature particularly difficult and inefficient. Overall, widespread use of the term is not indicative of a shared understanding of the construct.

### 4.2. Consensus on Definitions and Conceptualisations

Twenty-four articles ‘about’ food literacy developed definitions or reviewed existing definitions of the construct, while 51 different definitions of food literacy were cited in the 82 articles ‘about’ food literacy.

While the Vidgen and Gallegos [15] definition was the most commonly cited, new definitions of food literacy are constantly emerging, encompassing broader conceptualisations of the term. These definitions are usually developed as a result of scoping or systematic reviews of the existing literature and expert consensus that tend to differ from the definitions developed in consultations with the general public. The literature-based definitions tend to be broader in scope, encompassing constructs such as food security and food environments. This requires the general public to meet higher levels of knowledge, skills or behaviours and have a more critical and active understanding of the food system in order to be considered ‘food-literate’, placing a higher onus on the individual [38].

In an international consensus study, Fingland et al. [38] found that, while some international researchers believe food literacy should extend beyond what is proposed in the Vidgen and Gallegos [15] model, few disagree with the core domains and components of this conceptualisation (Figure A2). This provides a starting point for the development of international indicators of food literacy previously limited by inconsistent definitions and understandings of the construct across contexts.

We found no articles using the term ‘food literacy’ published by first authors based in low-income countries, which has been attributed to limited research budgets, low salaries, poor infrastructure and facilities and political instability [39]. However, since our search was conducted, articles from Ethiopia [40] and Uganda [41] have been published, further highlighting the relevance of the term.

Additionally, the recent COVID-19 pandemic further highlighted the relevance of food literacy and the role it plays in: (i) planning, selecting and preparing healthy meals [42,43]; (ii) empowering individuals, households, communities or nations to navigate the complex food environments and protect diet quality through change [44,45] and (iii) manage the planning and preparation of food even when financial circumstances change, which may alleviate food insecurity [46].

Overall, this review found the term ‘food literacy’ widely used and understood, and there is agreement on a core conceptualisation. Therefore, international scholars across all income levels that have engaged in food literacy research, identified by Fingland et al.’s [38] work and the present review, could provide valuable insight into developing indicators of food literacy. The development of a definition and international indicators endorsed by a peak international agency would be integral in significantly advancing and progressing the field of food literacy. With previous constructs, such as food security [47,48] and sustainable healthy diets [49], a United Nations (UN) definition supported by an agreed set of principals has allowed for national and global systems and monitoring. Further to this, food systems monitoring by the Food and Agriculture Organisation of the United Nations (FAO) [50] has identified consumer behaviours as a driver of the food system; however, there have been no measures reported for assessing food acquisition, preparation, meal practices and storage: all key components of food literacy. Therefore, the development of measures to assess components of the food system also relies on progressing international consensus and indicators.

### 4.3. Strengths and Limitations

The strengths of this research included adherence to the PRISMA-ScR guidelines and the broad scope of the review. The limitations include that this review was conducted up to 2019 and, therefore, did not describe the trajectory of food literacy during or post-COVID-19; however, a section was included to highlight relevant publications that have emerged since. The country of affiliation of the first author was not always the country where the research took place; this was a pragmatic decision due to the volume of articles coded but may particularly underrepresent countries where research is more often collaborative [39]. The article type was determined based on the journal’s classification of the article; as a result, there may be misclassifications of some article types; however, this was for pragmatic reasons. The In Cites categories were used to determine the disciplines of the papers, however not all journals were in this database which limited the analysis in this study. Therefore, a more robust discipline or area of study categorisation is needed. The articles that were ‘about’ food literacy were identified by inclusion of the term in the title, and while other methods were piloted, they were not easily replicable. Finally, the articles were requested from ResearchGate if they were not locally available; however, this, combined with the articles restricted to the English language, may have limited the generalisability of our findings.

## 5. Conclusions

This was the first comprehensive scoping review of the use of the term ‘food literacy’ within the English-language peer-reviewed literature. In total, 549 peer-reviewed journal articles, published over 21 years, were identified that used the term ‘food literacy’. The term has been used frequently and broadly throughout the literature over time, though there are inconsistencies in its application and definition. Agreement on a standardised definition of food literacy endorsed by a peak international agency is needed in order to progress the field.

## Figures and Tables

**Figure 1 nutrients-13-02006-f001:**
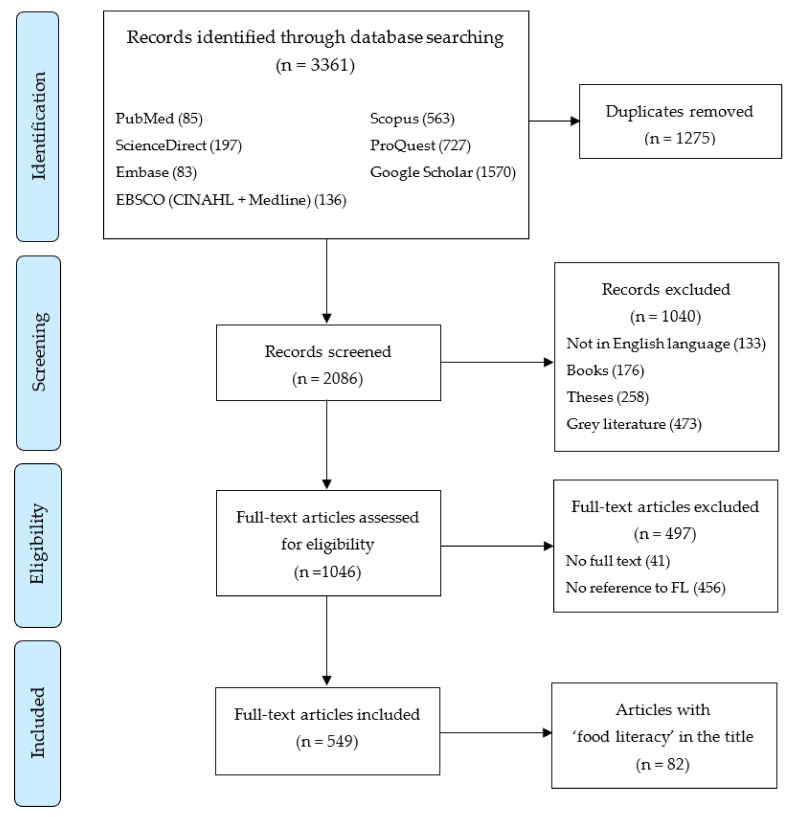
Prisma-ScR flow chart [30].

**Figure 2 nutrients-13-02006-f002:**
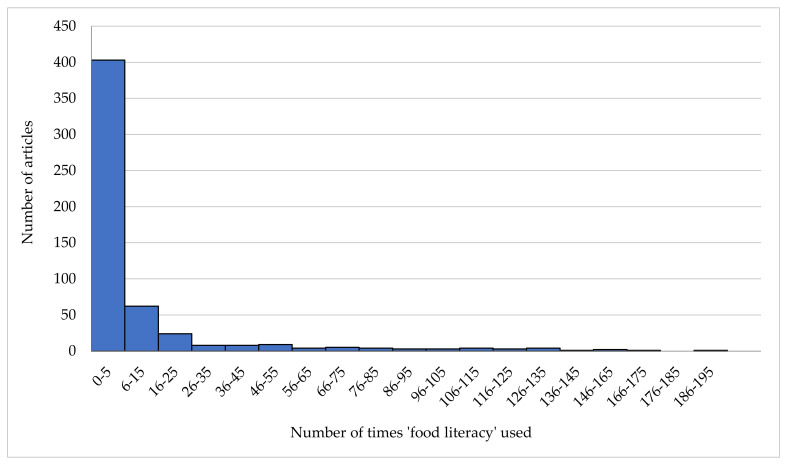
Frequency of the use of the term ‘food literacy’ within the articles containing the term (*n* = 549).

**Figure 3 nutrients-13-02006-f003:**
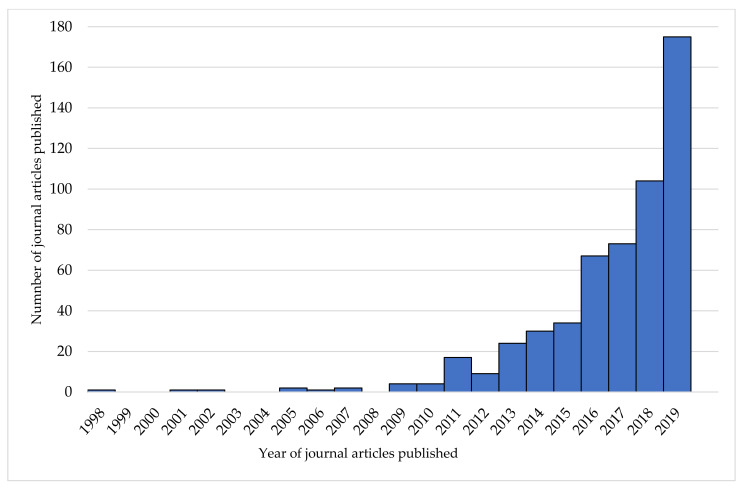
Trends in articles published using the term ‘food literacy’ from 1998–2019 (*n* = 549).

**Figure 4 nutrients-13-02006-f004:**
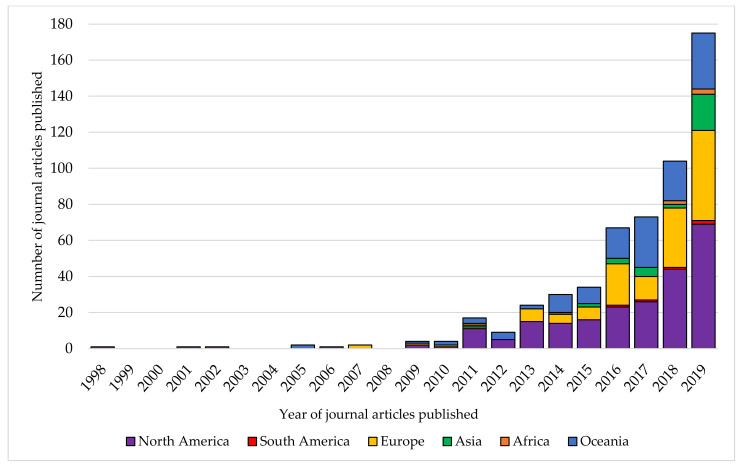
Trends in the continents of the journal articles published using the term ‘food literacy’ from 1998–2019 (*n* = 549).

**Figure 5 nutrients-13-02006-f005:**
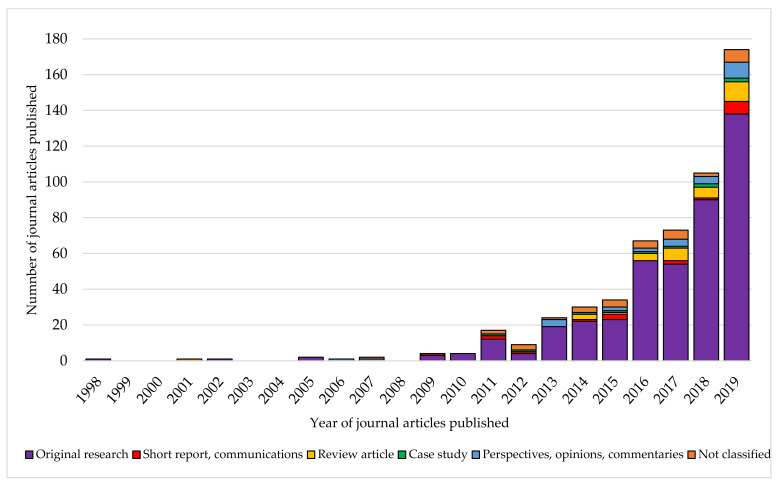
Trends in article types using the term ‘food literacy’ from 1998–2019 (*n* = 549).

**Figure 6 nutrients-13-02006-f006:**
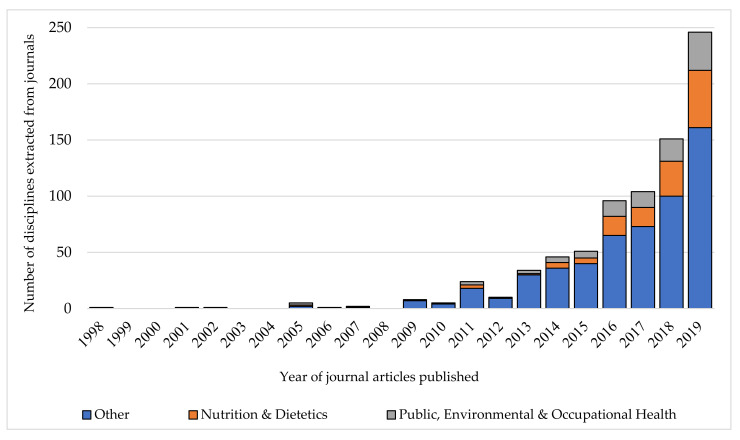
Trends in the disciplines of the articles published using the term ‘food literacy’ from 1998–2019 (*n* = 549).

**Table 1 nutrients-13-02006-t001:** Article types of all articles (*n* = 549) in comparison to articles primarily focused on food literacy (*n* = 82).

Article Type	Food Literacy-Focused Articles *n* = 82 (%)	All Articles *n* = 549 (%)
Original research	53 (65)	429 (78)
Review article	10 (12)	32 (6)
Perspectives, opinions and commentaries	6 (7)	27 (5)
Short report and communications	4 (5)	18 (3)
Case study	2 (2)	10 (2)
Not classified	7 (9)	33 (6)

**Table 2 nutrients-13-02006-t002:** A comparison of the aims of articles with a primary focus on food literacy.

Aim of Food Literacy Articles	Number of Articles in Each Category *n* = 82 (%)
Expert opinion articles ^1^	27 (33)
Definition articles ^2^	24 (29)
Intervention/program articles ^3^	17 (21)
Measurement articles ^4^	9 (11)
Observational articles ^5^	5 (6)

^1^ Conducted a study, usually with experts, or consulted the literature to inform future directions for food literacy research. ^2^ Developed a definition of food literacy or conducted a review of the existing definitions. ^3^ Conducted a study and assessed the food literacy as part of a program/intervention. ^4^ Developed a food literacy questionnaire or conducted a review of existing questionnaires. ^5^ Cross-sectional data collection on food literacy.

## Data Availability

The data presented in this study are openly available in the Appendix A associated with this publication.

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
