# Peer review of "Are We Closer to International Consensus on the Term ‘Food Literacy’? A Systematic Scoping Review of Its Use in the Academic Literature (1998–2019)"

_nutrients, 2021, doi:10.3390/nu13062006_

Round 1

Reviewer 1 Report

The manuscript concerns a systematic scoping review, aiming to portrait the use and development of the construct 'food literacy' within current scientific literature. Using PRISMA-ScR guidelines, the authors have composed an extensive data set of studies concerning 'food literacy' while summarily highlighting the type, country of origin, disciplines, and enumeration of term usage within the analysed articles. This study provides for a delimited illustration of current food literacy publications and its understanding.

Nonetheless, the manuscript requires improvement by reviewing a few minor to moderate concerns:

[1] The abstract would benefit of additional information regarding the conclusions of the study.

[2] Concerning spell check: (lines 51-52) were it is reviewers should state reviews; (line 130) the word were is missing on the paragraph 'Five-hundred and 129 forty-nine articles met the inclusion criteria and **were** included in subsequent analysis'.

[3] Within the manuscript Discussion, the authors affirm that food literacy is beginning to be more currently discussed in a broader spectrum within contexts of food security and food environments (in lines 346-347); however, no studies that integrate broader food-related contexts are referenced in this statement. Related to this concern, the authors also mention the importance of developing measures that integrate these broader contexts, particularly focusing on food systems (since consumers' behaviours is one of its main drivers), food security and sustainable diets, and urging for its assessment (e.g., lines 346-347; lines 390-404; line 424). 

Given the current global context of public health and its already acknowledged impact on food systems, food sustainability, and both food safety and food security by scientific research, the absence of literature developed during a complete year of COVID-19 pandemic seems to unfold as a severely significant handicap of this work by not include 2020 in its search, screening and selection strategies.

Making up for one of the two major concerns pointed out in this review, the authors are advise to briefly search for more recently developed research that acknowledges this singular public health reality, especially given its impact on food systems, consumers' food-related behaviours and, consequently, food literacy.

[4] As the second more significant concern and taking into account the authors' aim of systematically review definitions of 'food literacy', the lack of comparison among the definitions found weakens the impact of the study's purpose. As so, a comparison of the content of the most impactful definitions is also strongly advised.

Reviewer 2 Report

This is a systematic scoping review on “food literacy”, a term that is widely used in the field of nutrition, public health and behavior, to characterize/group populations and analyze accordingly.

This paper therefore is of interest since variations in food literacy definitions may affect comparative analysis, since study result heterogeneity increases.  

Pg 2, lines 49-51: need to be grammatically corrected “…what food literacy is and is not…” does not read well.  

Methods: one of the exclusion criteria was “(2) full text was not locally available”. What were the author’s actions in order to increase access to these 115 papers? The number is large and this could have influenced results.

The precise search terms are not provided. This is key in such studies, especially in terms of food literacy, since this term may have also been used in other ways. Please include in detail and/or add in supplementary file.

Also, it is to my opinion that any terms that is used to define food literacy should also be included.

Please explain in more detail why it was important to tabulate the number of times the term appeared in the text.  It may help to provide a table or graph for this.

As the authors have nicely reported, the term “food literacy” has been increased and is highly prevalent the past 5 years? Have the authors looked into definition consistency during this period specifically? It is not clear in the text.

How does the term differ by discipline? How important is it to have it clearly defined for all? Can it be differentiated by discipline based on the areas of study? I think this may be an option. Please discuss.

Reviewer 3 Report

It is an elegant and timely comprehensive scoping review of the use of the term ‘food literacy’ within the English-language peer-reviewed literature aimed at describing the frequency of use and reach of the term, identify changes in the use of the term over time, and describe how the term has been applied and defined within the literature. The authors concluded by assessing the need for an agreement on a standardized definition of food literacy endorsed by a peak international agency. 
